

# Hydro-Climatic Modelling of an Ungauged Basin in Kumasi, Ghana

Marian Amoakowaah Osei[1], Leonard Kofitse Amekudzi[1], David Dotse Wemegah[1], Kwasi Preko[1], Emmanuella Serwaa Gyawu[1], and Kwasi Obiri-Danso[1]

[1]Kwame Nkrumah University of Science and Technology, Ghana

*Correspondence to:* Marian A. Osei (marianosans92@gmail.com)

**Abstract.** The Owabi catchment which is about 69 km$^2$ provides about 20 % of water needs of the Kumasi metropolis has been in recent times prone to high anthropogenic activities that threaten water resource management. The Soil-Water-Assessment-Tool (SWAT) was used to assess the extent of these activities on the hydrology on the catchment from 1986 to 2015. Specifically, the model simulated historic and projected stream-flow and water balance. Initial results revealed the forest and topography played major role in water loss at the catchment as evapotranspiration and surface runoff were the dominant modulating pro-
cesses. Monthly calibration/validation of the model yielded satisfactory results with NSE (0.66/0.67), $R^2$ (0.67/0.67), PBIAS (8.2%/8.0%) and RSR (0.59/0.58). Nine sensitive parameters of which the catchment slope (CN2) ranked principal were found to control runoff amounts into the river. The model uncertainty was also quite low as the 95PPU enveloped about 50% of the observed streamflow within a width of 0.45 - 0.55. Furthermore, future streamflow predictions were modelled under RCP2.6,
RCP4.5 and RCP8.5 climatic scenarios, and two landuse scenarios, landuse category 1 and 2 (LU1 and LU2). An increasing trend of the downscaled rainfall totals between 2021 to 2050 for all RCPs were observed. This will positively impact stream-flow generation at the catchment under LU1. There is an expected deficit of streamflow amounts under LU2 relative to LU1, and a marginal reduction as compared to the baseline. In general, the model proved efficient in determining the hydrology parameters in the catchment and therefore has potential to be used for further modelling of water quality and pollution to aid
effective water resource decisions at the catchment.

## 1   Introduction

Rainfall plays a significant role in the hydrologic cycle and is an essential resource for global socioeconomic activities. Major stakeholders depend on various aspects of the cycle on different time scales. For instance, rain-fed agriculture (dominant in Africa) has been found to operate well if soil moisture is replenished at least every 15 days. Stream-flow which is also important
for flood control, hydro-power, navigation and ecological factors has its high and low extremes being controlled by rainfall and groundwater base-flow respectively (Lofgren and Gronewold, 2014).

Water planners and managers rely on assumptions that past and future climatic trends will be the same and hence, water supply systems such as dams are built with these assumptions in mind (Mukheibir, 2007). However, current climate change



and variability is offsetting precipitation globally, posing a dire threat to water resource management (WRM). According to studies by Kankam-Yeboah et al. (2012) [and see references therein], without climate change considerations, Ghana is likely to be water deficient by 2025. This situation is projected to exacerbate with increased anthropogenic activities, which will negatively impact future water resources by restricting their use to meet growing demand (Kankam-Yeboah et al., 2012).

WRM problems have been tackled by many countries through the Integrated Water Resource Management (IWRM) initiative. IWRM presents a holistic approach in water management by providing a stronger coordination between stakeholders in planning and managing water resources in river basins (Zhang et al., 2014). Ghana's implementation of IWRM at the river basin level begun with the most 'water stressed' basins in the country (Water Resource Commission, 2012). One of such basins is the Pra of which the Owabi catchment serves as a sub-tributary. The dam within the catchment was designed to produce 20%

of total potable water requirement of the Kumasi metropolis and its environs. However, increasing human activities including deforestation, sandwinning, illegal logging, indiscriminate dumping of waste has reduced the quality and quantity of water production at the dam. Consequently, this has imposed severe threat on water resource and ecosystem mangement. In order to understand the impacts of human activities on the different hydrological processes, a hydro-climatic modelling assessment was carried out. A well-documented and tested hydrological model which has been widely used for these assessments is the

Soil-Water-Assessment-Tool (SWAT) (Arnold et al., 2012a, b; Shope et al., 2014).

The SWAT model incorporates a geographical information system (GIS) interface to give meaningful insights into the water balance, sediments and pollutant transfer in a drainage network (Uzeika et al., 2011). The SWAT model is prominent for its continuous long-term simulations of hydro-climatic variables (Sudjarit et al., 2015) as well as satisfying its developmental aim of testing and predicting water and sediment routing in ungauged basins (Gayathri et al., 2015). It is capable of evaluating the

effects of best management practices on water resources in both large and small river basins [see (Uzeika et al., 2011; Sudjarit et al., 2015; Shope et al., 2014; Me et al., 2015)] and has returned favourable performance rate when calibrated.

For instance, Abraham et al. (2007) calibrated and validated the SWAT model for an Ethiopian watershed and found slight under-and over-estimation of peak flows for some years. Nonetheless, the overall performance of the model was good for watershed simulations. Govender and Everson (2005) used the manual calibration technique to model stream-flow for two

catchments in South Africa where the increasing demand for timber has drastically changed land use/cover and hydrological processes. Although there was a good response between simulated and observed stream-flows, the model was unable to account for evapotranspiration losses. Schuol and Abbaspour (2006) deduced that the model has a huge potential for freshwater quantification after application to a four million $km^2$ West African catchment. A study by Begou et al. (2016) at the Bani catchment revealed that calibration at the subbasin scale resulted in better performance than using global parameter set. Furthermore,

the model performed well on daily and monthly scales with a good predictive uncertainty, but sometimes highly overestimate potential evapotranspiration.

The aim of this study therefore, was to model the hydro-climate of the Owabi catchment. Specifically, we utilised the SWAT model to simulate stream-flow and establish the water balance for the catchment. The influence of parameter settings on stream-flow was determined through sensitivity/uncertainty analysis; calibration and validation were also performed using

SWAT-CUP (SWAT Calibration and Uncertainty Prediction) tool. Finally, we used the SWAT model to forecast the streamflow



for the catchment under different climatic and landuse scenarios. The remaining part of this paper is structured as follows; section 2 presents the methodology, results and discussions in section 3 and conclusion given in section 4.

## 2 Methodology

### 2.1 Study Site

5   The Owabi catchment (Figure 1) has been designated since 1988 as the only inland Ramsar site in Ghana. redIt comprises of the forest reserve (sanctuary) and the Owabi waterworks and covers about 69 km² (Akoto and Abankwa, 2014) of land area. Its location is between latitudes 6.7292° N and 6.7519° N and longitudes 1.7139° W and 1.6704° W. Within the catchment is a 13 km² forest reserve enclosing a water reservoir (Commission, 2014).This forest reserve is one of the smallest conservation sites in Ghana and its protection responsibility lies solely with the Department of Game and Wildlife (Adubofour, 2011).

10   Nonetheless, protection of the site has not deterred high human encroachment and other illegal activities. The hydrological unit is situated in the inner perimeter of the sanctuary (Figure 1). The river was dammed in 1928 with the primary aim of supplying 20% of potable drinking water to the Kumasi metropolis.

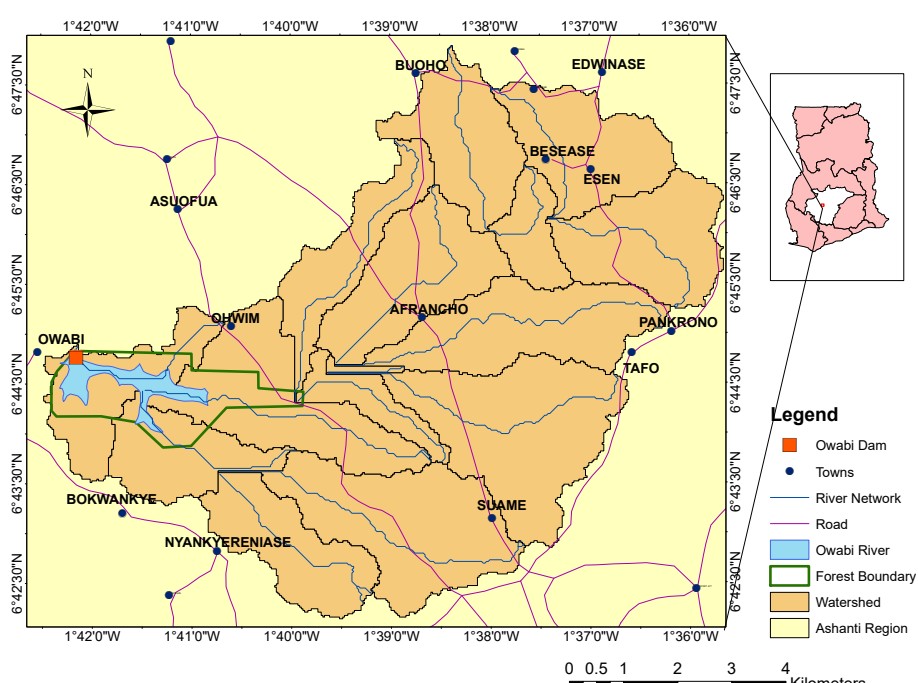

**Figure 1.** The Owabi catchment comprising of the Owabi Dam and the Forest Reserve

    The catchment rests in the forest belt of Ghana where rainfall is strongly modulated by the West African Monsoon (WAM) and convective activites resulting from movements of the Intertropical Discontinuity (ITD). The WAM is primarily driven by



temperature and energy gradients between the Gulf of Guinea and the Sahara (Amekudzi et al., 2015). Movement of the ITD results in bimodal rainfall regimes in this region with the presence of the river and dense vegetation inducing a microclimate which influences the rainfall patterns, temperature and humidity in the catchment. The mean rainfall is about 1450 mm per annum (Aryee et al., 2017) and average monthly temperatures between 24.6 - 27.8° C. Geologically, the area falls within

the Birimian meta-sediment of the Kumasi Basin which consists of phyllites, granodiorites, schists, greywackes, tuffs and the associated granitoid (Adubofour, 2011; Commission, 2014) and Orthic Acrisols as the dominant soil type.

## 2.2 SWAT model description

SWAT is a river-basin scale semi-distributed and physically based model that operates on a daily timescale. It was initially developed by the United States Department of Agriculture to model land management practices on sediment, water and agro-

chemical yields in ungauged catchments. The model is highly rated for its computational efficiency and long term continuous simulations. The important inputs of the model include; rainfall and temperature, digital elevation model, soil and land use maps as well as output simulations of water balance, nutrient and sediment loadings (Arnold et al., 2012b).

The computational hydrology framework of the model is based on the land and water/routing sections of the hydrological cycle. The land division regulates the amount of water, nutrient, sediment and pest loadings into the main channel in every

subbasin. The latter division defines the path of water and sediments through the basin channels to the outlet (Sudjarit et al., 2015). The initial step for watershed simulation is the watershed delineation where subbasins and hydrological response units (HRUs) are defined (Arnold et al., 2012a).

SWAT utilises Equation 1 to simulate the hydrological cycle (Ghoraba, 2015). The hydrological cycle is climate dependent and supplies energy and moisture inputs such as daily rainfall, maximum and minimum air temperatures, wind speed, relative

humidity and solar radiation. The data can be read directly from files by SWAT to produce simulated data at runtime.

$$SW_t = SW_o + \sum_{i=1}^{t} (R_{day} - Q_{surf} - E_a - w_{seep} - Q_{gw})_i \tag{1}$$

where t is time (days), $SW_o$ and $SW_t$ are the initial and final soil water content, $R_{day}$, $Q_{surf}$, $E_a$, $w_{seep}$ and $Q_{gw}$ are the quantities of rainfall, surface runoff, total evapotranspiration and return flow respectively. All parameters have units in $\mathrm{mm}$ and $i$ represents the parameter value for a day.

## 2.3 Data Sources

### 2.3.1 GIS Interface and Spatial Dataset

The open source QGIS version 2.6.1 and QSWAT version 1.4 (SWAT v2012), downloadable from http://swat.tamu.edu/ software/, were employed for the study. Spatial input dataset were obtained from various open source websites as shown in Table 1. The catchment elevation (Figure 2) was within 220 m and 394 m above mean sea level. All these data excluding

hydro-meteorological parameters were in a raster format and geoprocessed for model input as described in Dile et al. (2016).





**Table 1.** SWAT input database

| Data Type | Description | Available Sources | Spatial resolution |
|---|---|---|---|
| Digital Elevation Model | SRTM 1-Arc-Second Global v3 | www.earthexplorer.usgs.gov | 30 m×30 m |
| Land-Use Raster | European Space Agency annual 1992-2015 | UCLouvain (2017) | 300 m×300 m |
| Soil Raster | Digital global soil map FAOv3.6 | WaterBase | 1:5000000 m |
| Historic meteorological data (1980-2015) | Daily rainfall and temperature | Ghana Meteorological Agency | station point data |
| Climate Projection data (2020 - 2050) | Daily rainfall and temperature | CCCMA | 0.44°×0.44° |
| Hydrological data (2001-2010) | Monthly streamflow | Ghana Hydrological Service Department (Offin River) | station point data |

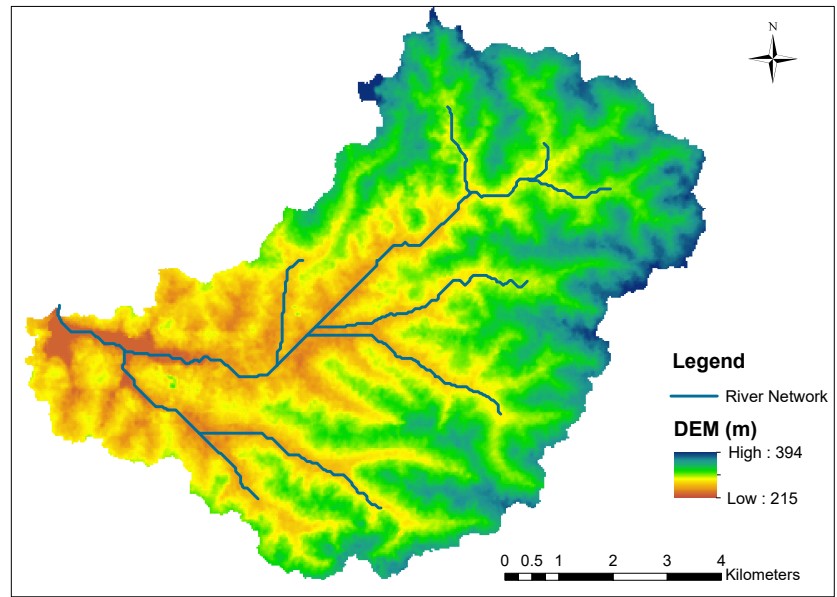

**Figure 2.** Digital Elevation Model for the Owabi Catchment

After model run, a total area of 69.72 km$^2$ basin boundary was delineated, with 16 subbasins and 80 hrus. Six dominant land-use categories and one soil type were observed within the Owabi catchment as shown in Figure 3 and Table 2. The observed soil (Orthic Acrisols) are deeply weathered and consist of thin, dark greyish brown, humus-stained, sandy loam and silt loam topsoils with moderate fine granular structure.



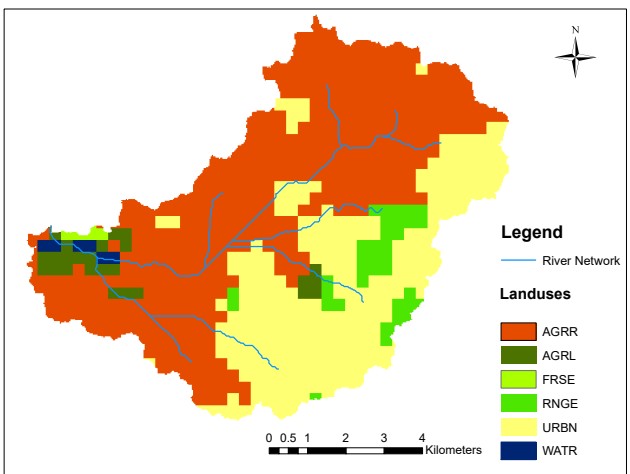
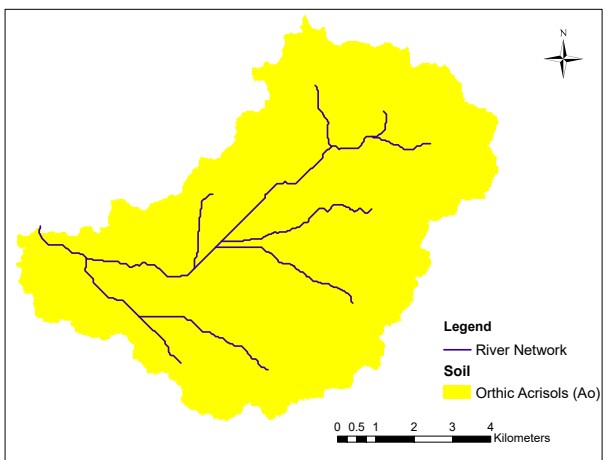

**Figure 3.** Land use categories (left) and soil type (right) for the catchment

**Table 2.** Land-use and soil categories within the Owabi catchment

| Description | QSWAT Code | Percentage within Watershed |
|---|---|---|
| Grassland | RNGE | 4.68 |
| Cropland | AGRR | 57.19 |
| Evergreen Forest | FRSE | 0.39 |
| Water body | WATR | 0.81 |
| Natural Vegetation | AGRL | 3.33 |
| Urban/Settlement | URBN | 33.60 |
| Orthic Acrisols (soil) | Ao1-ab | 100 |

### 2.3.2 Hydro-Meteorological Input

Daily rainfall and temperature (minimum and maximum) were obtained from Ghana Meteorological Agency (GMet) from 1980 to 2015 (Table 1). Rainfall and temperature gaps were filled using the arithmetic averaging method from the closest meteorological stations. These stations included, Barekese, Offinso and Kumasi Airport.

5    The paucity of stream-flow data was a major limitation of this paper for model calibration and validation. In the decade-long review of the prediction in ungauged basins, it has been revealed that regionalisation and other genetic networks can be used for stream-flow determination (Hrachowitz et al., 2013). Since streamflow records were available for the nearest catchment, the Offin River, which is about 11 km from Owabi catchment, we implemented the spatial proximity global arithmetic mean





method as used in Oudin et al. (2008); Swain and Patra (2017); Arsenault and Brissette (2016) and references therein. Thorough review of these methods can be found in Hrachowitz et al. (2013).

### 2.3.3 Projection Dataset

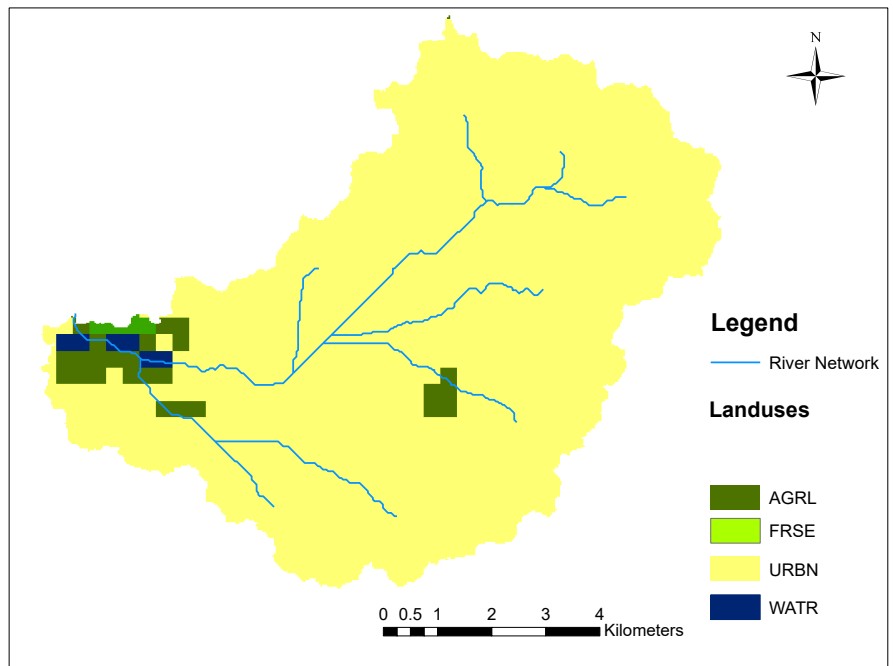

**Figure 4.** Projected landuse category (LU2)

A 30-year (2021-2050) streamflow for the Owabi catchment was predicted under two land-uses and three climate change scenarios. For landuse scenario 1 (LU1), we considered unchanging landuse categories as observed in the baseline trend. In land-use scenario 2 (LU2) (Figure 4), we assumed the conversion of all landuses except forest (FRSE), natural vegetation (AGRL) and water (WATR) categories into settlement/urbanisation areas.

Future daily rainfall and temperature (minimum and maximum) from the Canadian Regional Climate Model were used as climate forcings for the models. These were projected under three Representative Concentration Pathways (RCP2.6, 4.5 and 8.5). The climatic projection data were bias corrected using the distributive mapping (statistical downscaling) technique in the CMhyd (Rathjens et al., 2016) tool. Among other bias correction methods, the distributive mapping was employed due to its assumption of gaussian distribution of the datasets, as well as its reliability from previous studies as reviewed in Teutschbein and Seibert (2012).





**Table 3.** Statistical Indices and their optimal thresholds for stream-flow according to Moriasi et al. (2007) and Abbaspour (2015)

| Objective function | Threshold (for stream-flow) |
|---|---|
| p-factor | $\geq 0.70$ |
| r-factor | closer to 0 |
| t-stat | larger absolute value |
| p-value | $\leq 0.50$ |
| NSE | $\geq 0.50$ |
| $R^2$ | $\geq 0.50$ |
| PBIAS | $\pm 25\%$ |
| RSR | $\leq 0.70$ |

## 2.4 Calibration, Validation and Uncertainty Assessment

The autocalibration tool, SWAT-CUP with the Sequential Uncertainty Fitting version 2 (SUFI-2) algorithm was used for cali-
bration techniques. SUFI-2 is known to estimate both parameter and model uncertainties in hydrological models (Abbaspour,
2015). Table 3 shows the objective functions used in assessing the overall model performance as well as their required satisfac-
tory thresholds as described in Moriasi et al. (2007) and Abbaspour (2015). The Global sensitivity test for sensitivity analysis
was evaluated using the t-stat and the p-value whiles p and r factors were used for uncertainty analysis at a 95% prediction
uncertainty (95PPU). The others included; Nash-Sutcliffe Efficiency (NSE), Percentage Bias (PBIAS) and RMSE Standard
Deviation Ratio (RSR) [see Appendix A for equation details].

## 3 Results and Discussions

## 3.1 Water Balance

According to Ghoraba (2015) [and references therein], the most important components of the water balance of any hydrological
basin are the rainfall, surface runoff, baseflow, lateral flow and evapotranspiration. Unlike rainfall that is easily measureable,
the other variables need prediction for their quantification. The SWAT model was run for 36 years, with the first 6 years used
as model initialisation (warm-up). The 30-year seasonal water balance for the catchment is visualised in Figure 5, which shows
that, the three most dominant pathways for water loss at the catchment are evapotranspiration, surface runoff and percolation
to shallow aquifer (in descending order). Due to the quick response to rain and a hydrological soil group of D within the entire
basin, surface runoff (Hortonian) contributes to about 75% of streamflow generation whiles the latter accounts for 44% of the
total variance in rainfall amounts. Loss of water to deep aquifer was minimal with a ratio of 0.01.

   The 30-year potential evapotranspiration (PET) average as calculated from the Hargreaves method, exceeded mean rainfall
amounts by about 6.36%. This can be attributed to the distance of the catchment to the equator and hence much higher insolation
and temperatures that favours the potential evapotranspiration (Sanderson et al., 2010). Furthermore, as stated earlier, the



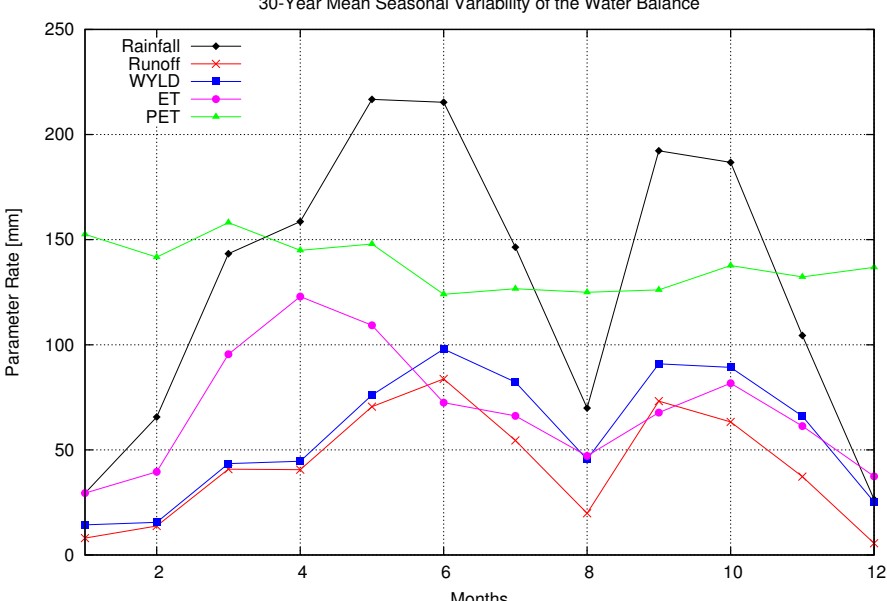

**Figure 5.** A 30-year (1986-2015) mean monthly water balance for the Owabi catchment

catchment includes a 13 $km^2$ tropical rain forests with evergreen decidous and semi-decidous trees which have been known to be sites for potentially high evapotranspiration rates (Sanderson et al., 2010; Kumagai et al., 2005; Bonal et al., 2016). For the seasonal trends, higher PET was observed during the onset of the rains in March, but this begins to reduce when penetrating into the major and minor wet seasons. This can be linked with the rigorous convective activities that occur due to the northward

movement of the ITD in early March. During the wet season, PETs are low and constant from June to September with slight increases for the dry season. The former condition might arise from high humidity and thick monsoonal clouds that negatively affect the rate of PET by reflecting most insolation to the land surface.

The Owabi catchment falls under a wet climatic regime which is known to support higher evapotranspiration with the presence of dense forested vegetation. These forests enhance evaporation by assisting in the aerodynamic transport of intercepted

water into the atmosphere (Legesse et al., 2003). The actual evapotranspiration (ET) rates were highest during the northern hemisphere summer monsoon especially July, also in response to convective activities. The contrary effect can be observed for the dry months, where high temperatures and quick evapotranspiration are likely to leave the surface soil depleted of moisture (Sharma, 1988). Any potential evaporation would occur from groundwater storage through the root-stem-leaf of trees and most plants especially grasslands and natural vegetation could reach their wilting point. Circumstances like these are likely to give

rise to hydrological drought within the catchment. On the other hand, the root nature of the forested trees could reach more depths for soil water in order to maintan transpiration for the dry period and subsequently result in a higher net average annual evaporation.





Lateral flow played quite an insignificant role in the water budget with values consistently constant regardless of the season. Water yield (surface runoff + subsurface flow) is an essential component of the hydrological cycle and determines how much water leaves the outlet at a given time (Ghoraba, 2015). The rate of water yield was quite high at the catchment which favours the Owabi water treatment plant and other agricultural activities. However, its seasonal fluctuation is controlled by groundwater

(return flow) in the dry season and overland flow (surface runoff) in the wet period. Thus, although there may be little to no rains within the catchment in the dry season, return flow increases the water yield whiles the surface runoff was the modulator in the wet season (Ogden et al., 2013; Miguez-Macho and Fan, 2012; Dralle et al., 2016). Groundwater contribution to streamflow during the dry season has been studied by Ogden et al. (2013) [and references therein] as the "sponge effect". The sponge effect assumes that well developed forests such as the Owabi forest reserve promote high infiltration and groundwater recharge

during the wet season leading to increased streamflow during the dry season. The eight month of August is termed the little dry spell; wind directions are still southwesterly, however, the ITD is displaced northward and this does not favour rains in the south of the country. The processes are observed to decline but not as much as during the major dry season.

## 3.2 Calibration Analysis

### 3.2.1 Sensitivity Analysis

For calibration analysis, 14 parameters (see table in Appendix 5) that affect surface runoff and baseflow have been selected from literature. The results of the global senstivity analysis showed that nine (9) of these parameters regulated the outlet streamflow in the basin. These included surface (CN2, SURLAG), soil (ESCO, SOL_BD, SOL_AWC), channel (CH_N2) and groundwater (ALPHA_BF, RCHRG_DP, GW_REVAP) processes. Almost all sensitivity studies using SWAT-CUP have obtained CN2 to be the most sensitive parameter of all watershed analysis [see Arnold et al. (2012b) and references therein]. In the present study,

CN2 also ranked as the principal most sensitive parameter, which is concurrent with most findings of hilly to mountainous terrains. The catchment topography ranged from flat to hilly with slope values between 69% (minimum) to 82% (maximum).

### 3.2.2 Monthly Calibration and Validation

The streamflow data obtained from the Offinso basin contained data between 2001 to 2010. As such, the model was calibrated from 2001 to 2006 (6 years) and validated from 2007 to 2010 (4 years). The plot of the calibrated and validated model is shown

in Figure 6.

During calibration, the model was consistent in simulating most of the seasonal streamflow dynamics and concurrently followed the the rainfall pattern. The parameter uncertainty that results from non-uniqueness of effective model parameters, conceptual model and input uncertainties (Schuol and Abbaspour, 2006) were satisfactory as observed from 95PPU (Table 4). The model bracketted about 51% of the observed data at the expense of a quite low enveloping width 0.55. The model uncer-

tainties would be the inability to incorporate some important processes occcuring within the catchment in the SWAT model setup.





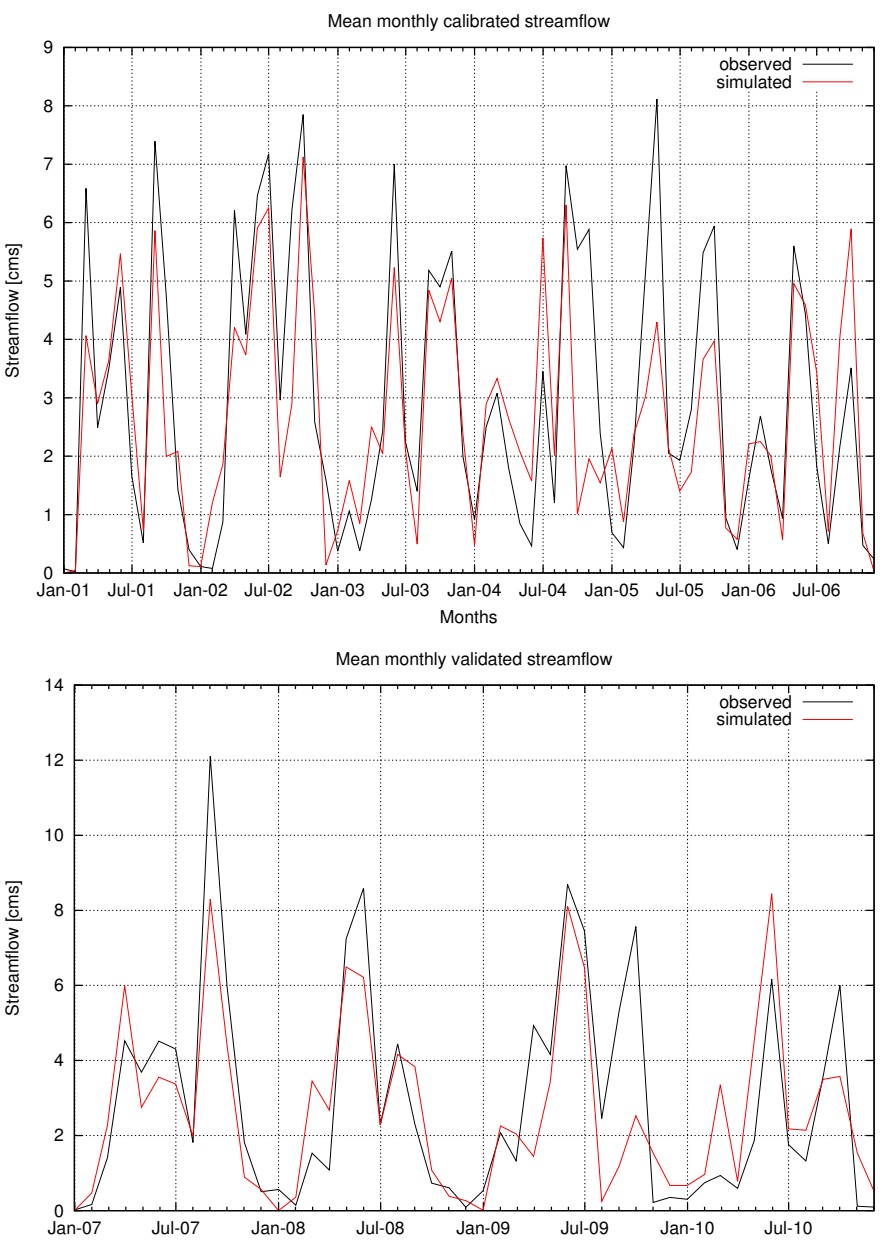

**Figure 6.** Mean monthly simulated stream-flow for calibration [top: (2001 - 2006)] and the validation period [bottom: (2007 - 2010)]

The primary observational data uncertainty at the catchment arises from the point measured rainfall data. These may include:
(1) Systematic errors associated with the rainfall measurements from the Owabi catchment. It has been ascertained that the
Owabi meteorological station has not been relocated since the beginning of the time period used for this study. Hence any



errors may arise from faulty gauges or the inability of the gauge to measure intense short and long duration of storm events. (2) The inherent errors associated with the neighbouring rain gauges around the Owabi catchment, which could also propagate errors through the computational framework of the data gaps in-filling method to the model run. Point scale rainfall uncertainty usually has average systematic error magnitudes of 5-6% and random error magnitude of about 5% (McMillan et al., 2012).

Secondly, in the application of the spatial proximity regionalisation scheme, it has been deducted from studies that for highly satisfactory results, there should be a dense stream gauge network in the vicinity of the proposed catchment (Hrachowitz et al., 2013). The Owabi catchment falls short of this criteria with the closest gauging station found about 11 km from the catchment area. Therefore the errors uprising from the regionalisation of streamflow is likely to be transferred directly to the calibration process, especially in the simulation of extreme event such as high peak flows. The stage-discharge measurement of streamflow

at the Offin River could also introduce errors in the discharge calculation, as associated in both instrumentation and quality control which results in potential outliers in dataset.

The model showed a satisfactory performance, as evident in the NSE (0.66) and $R^2$ (0.67) considering that the observed data was obtained from a catchment which is about 12 times the size of the study area. The models' performance was enhanced especially during the dry seasons with consistently slight overestimation of streamflow. Groundwater parameters such as al-

pha baseflow, groundwater recharge depth and revaporation of groundwater become extremely important in low flow periods during dry seasons at the catchment to support water production by the Owabi water processing plant. Although groundwater delay is a medium sensitive parameter and has a low index on the long-term, its delay period ($\approx$ 82 days) is appropriate for recharge flow for the dry season. It was observed also that for rainy months (see Figure 7) in which the mean monthly rainfall amounts were lower (hence low streamflow amounts), the model treated such periods as dry and always over-predicted the

streamflow amounts. This is observable in March to July 2004 and September to October 2006, where streamflow observed for the rainy period was lower than other calibration years. The low rains experienced in the former months resulted in short intense rainfall in September to December 2004, which was poorly captured by the model. Overall, the prediction error for streamflow calibration was within an acceptable range as reflected in the PBIAS (8.2%) and RSR (0.59%).

The monthly validation (Figure 6) also showed a good output between the simulated and the observed streamflow datasets.

The overall model performance conforms with a hydrological review undertaken by Gassman et al. (2007). The highest observed streamflow peak during calibration was obtained in May 2005 (8.116 cms), however this was poorly simulated by the model (4.299 cms). Alternatively the highest best simulated peak streamflow by the model was in October 2002 (7.124 cms). Likewise, for validation, the peak was seen in September 2009 for observed (12.105 cms) and simulated (8.302 cms). Table 4 shows a summary of the statistics obtained for model calibration and validation.

## 3.3   Hydro-Climatic Projections

### 3.4   Rainfall Projection Trends

As observed in Figure 8, rainfall is expected to increase for the respective concentration pathways for the next 30 years (2021-2050). The 30-year averages were 1392.39 mm, 1450.09 mm and 1354.74 mm which represented approximately 9%, 14% and



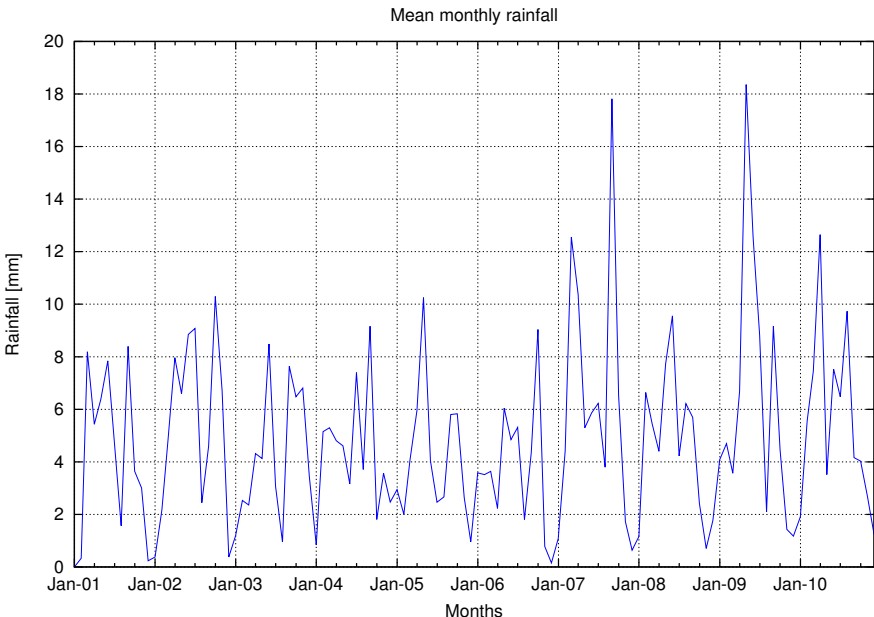

**Figure 7.** Mean monthly rainfall timeseries for the calibration/validation period (2001-2010)

**Table 4.** Summary of calibration and validation statistics

| Objective function | Calibration (2001-2006) | Validation (2007-2010) |
|---|---|---|
| p-factor | 0.51 | 0.52 |
| r-factor | 0.55 | 0.45 |
| NSE | 0.66 | 0.67 |
| $R^2$ | 0.67 | 0.67 |
| PBIAS | 8.2% | 8.0% |
| RSR | 0.59 | 0.58 |
| Mean obs | 2.90 | 2.90 |
| Mean sim | 2.66 | 2.66 |
| Stddev obs | 2.30 | 2.27 |
| Stddev sim | 1.83 | 2.85 |

6% increases for RCP2.6, RCP4.5 and RCP8.5 as against the baseline (1986-2015) average of 1274.05 mm. Rainfall peaks are expected to occur in 2039 (RCP2.6), 2027 (RCP4.5) and 2024 (RCP8.5). Rain intensity and rainy days are expected to increase as projected by the IPCC (Pachauri et al., 2014) for monsoonal regions, with the highest amounts probably under RCP4.5. In addition, observations from the daily rainfall suggests that the wet season will continue to get wetter whiles the dry season

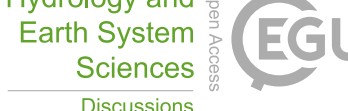



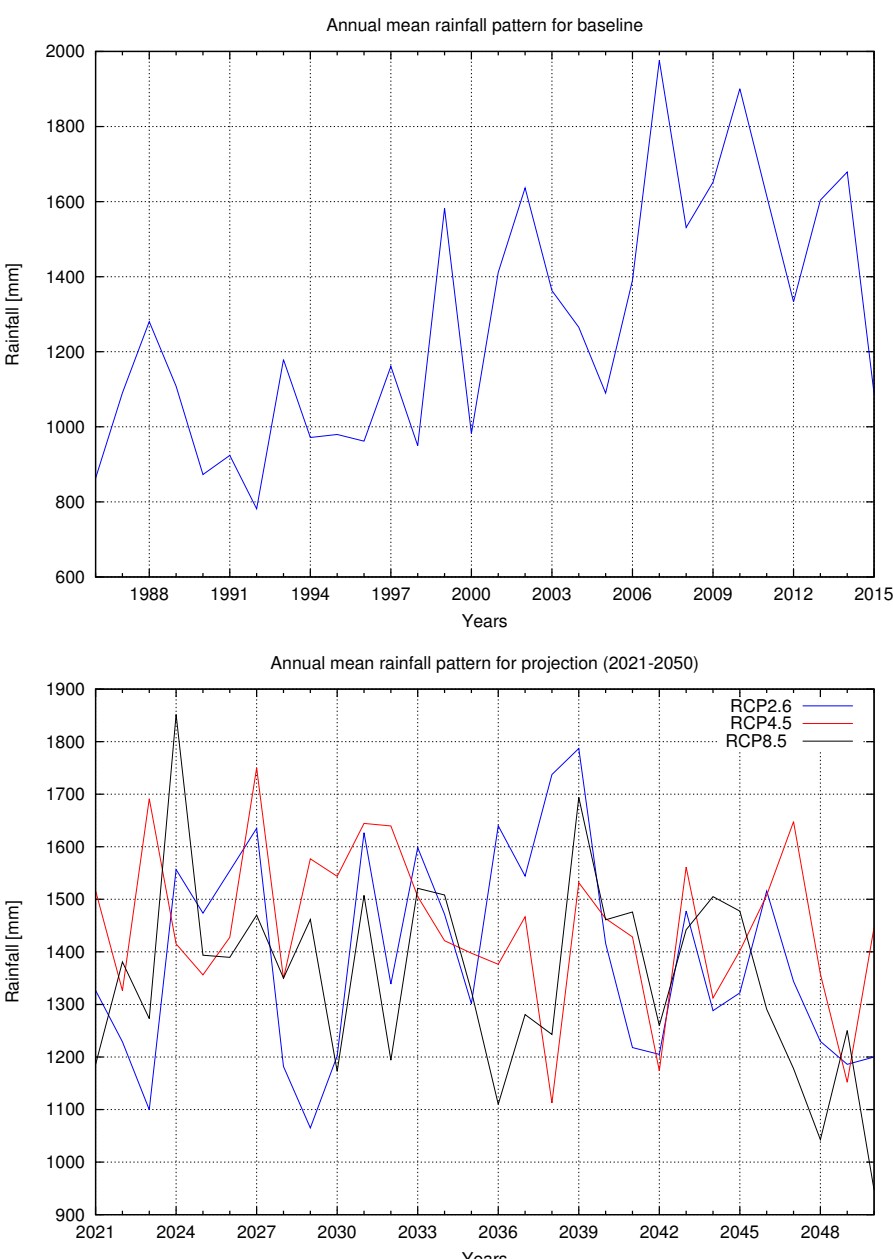

**Figure 8.** 30-year annual mean rainfall for projection [top: (1986 - 2015)] and baseline [bottom: (2021 - 2050)]

gets drier as evident from consistent zero mean monthly values for the dry period. The increase in rainfall further implies that, under LU1, actual evapotranspiration still remains the dominant mode in which water will be lost at the catchment. Although this might be welcoming for water resource management, the likely disadvantage is the reduction in water quality due to high





sediment load from increased runoff. This could ultimately stress water production for the Kumasi metropolis. Hence, this wake-up call should bring together water planning stakeholders for the implementation of appropriate measures to meet future water demands.

## 3.5 Streamflow Projection Trends

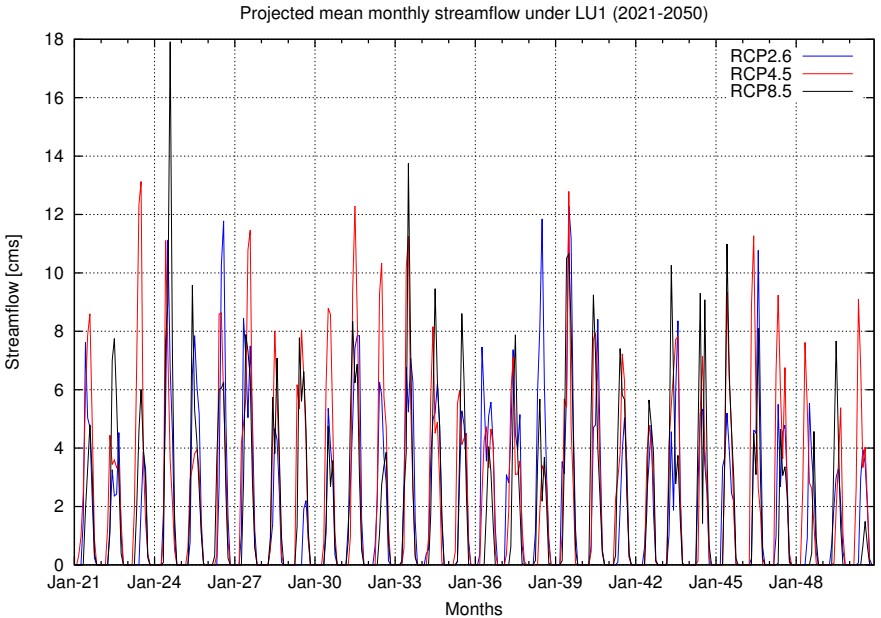

**Figure 9.** Landuse scenario 1 (LU1) stream-flow forecast

5    The calibrated model was used to generate streamflow for LU1 and LU2 scenarios. In response to high frequency and intensity of future rainfall (Figure 8), streamflow quantities are likely to increase under all RCPs in LU1 (see Figure 9). If this increase is properly harnessed, water production could by boosted for both socio-economic and agro-irrigational demands. On the other hand, streamflow trends are expected to decrease under LU2 (Figure 10) in response to increases in settlement, but the discharge trends are likely to show less dynamics to the baseline amounts (Figure 6). The alteration in the catchment hydrology under LU2 shows a significant streamflow deficit of about 42.6% (RCP2.6), 43.35% (RCP4.5) and 41.33% (RCP8.5) relative to LU1. This could likely impact the availability and amount of water for storage and production to meet local demands. Although under all land scenarios, zero monthly discharge was observed, we assume that under proper management conditions, there would be optimum storage of water in the reservoir throughout the dry season. The projection streamflow peaks is likely to be independent of the rainfall peaks observed in Figure 8.





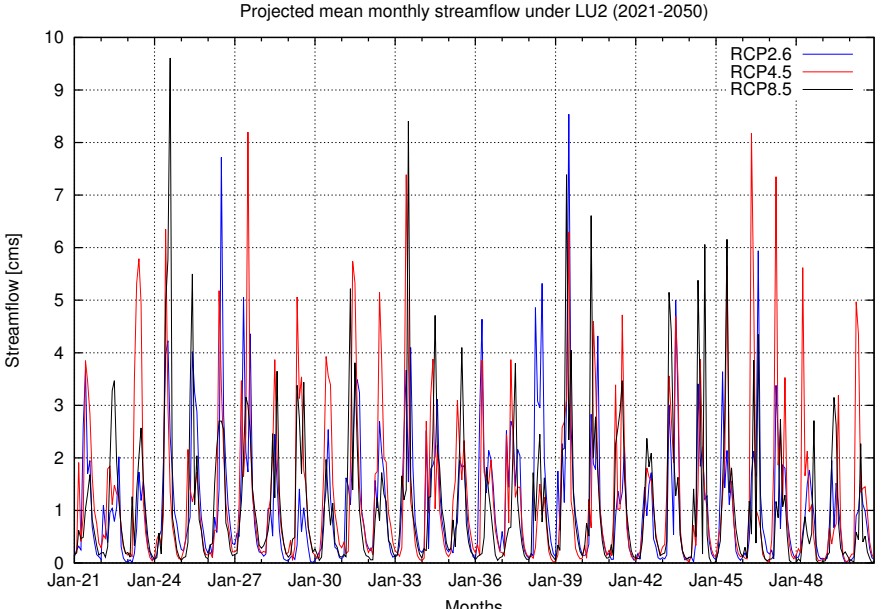

**Figure 10.** Landuse scenario 2 (LU2) stream-flow forecast

## 4  Conclusions

The Soil-Water-Assessment-Tool (SWAT) has been used for hydro-climatic assessment of the Owabi catchment from 1986 to 2015. Specifically, the model simulated historic and projected stream-flow, water balance, as well as model calibration and validation for the catchment. QGISv2.6 interface was used to launch SWATv2012 for QSWAT v1.4. Initial results revealed the
forest and topography played major role in water loss at the catchment as evapotranspiration and surface runoff were the most dominant modulating processes. The SUFI-2 algorithm within SWAT-CUP was used for calibration purposes on the monthly temporal scale. The sensitivity analyses showed nine (9) parameters of which the catchment slope (CN2) ranked first as the most sensitive in controlling runoff amounts into the river. Overall, the SWAT model revealed a satisfactory performance which was evident in the statistical indices used for both calibration and validation. The uncertainty was also quite low as the model
enveloped between about 50% of the observed streamflow within a width of 0.45 - 0.55. Future streamflow predictions were modelled under RCP2.6, RCP4.5 and RCP8.5 climatic scenarios and two landuse scenarios, LU1 and LU2. The downscaled rainfall trends showed increases in rainfall totals between 2021 to 2050 for all RCPs. This will positively impact streamflow generation at the catchment under LU1. With respect to streamflow amounts under LU2, there will be a likely deficit relative to LU1, and a marginal reduction as compared to the 10-year calibrated/validated baseline. In general, the use of the SWAT
model for hydrological assessment of the Owabi catchment has been successful and further studies on the assessment of water quality and pollution is currently being undertaken to provide a holistic view of water resource management at the catchment. This would in the long-term aid effective decision making and boost water production for the Kumasi metropolis.





**Appendix A: Equations of Objective Functions**

$$NSE = 1 - \frac{\sum_i (Q_m - Q_s)_i^2}{\sum_i (Q_{m,i} - \bar{Q_m})^2} \tag{A1}$$

$$R^2 = \frac{\left[\sum_i (Q_{m,i} - \bar{Q_m})(Q_{s,i} - \bar{Q_s})\right]^2}{\sum_i (Q_{m,i} - \bar{Q_m})^2 \sum_i (Q_{s,i} - \bar{Q_s})^2} \tag{A2}$$

$$PBIAS = 100 \times \frac{\sum_{i=1}^{n}(Q_m - Q_s)_i}{\sum_{i=1}^{n} Q_{m,i}} \tag{A3}$$

$$5 \quad RSR = \frac{\sqrt{\sum_{i=1}^{n}(Q_m - Q_s)_i^2}}{\sqrt{\sum_{i=1}^{n}(Q_{m,i} - \bar{Q_m})^2}} \tag{A4}$$

where Q is the discharge, $m$ and $s$ are measured and simulated parameters, bar is the average variable and $i$ is the $i^{th}$ observed or simulated data.

**Appendix B: Description of parameters**

*Author contributions.* The study was designed by L.K. Amekudzi, D.D. Wemegah and carried out by M.A. Osei. Writeup and data analysis
10   was done by M.A. Osei and proofreading by all authors.

*Competing interests.* The authors declare no conflict of interest.

*Acknowledgements.* We express our sincere gratitude to Building Stronger Universities Phase 2 (BSU II) and Kwame Nkrumah University
of Science and Technology (KNUST) for funding of this study. Mrs. Jamilatou Begou for our training in the use of the SWAT model and Mr.
Charles Yorke for making available the climate data from GMet.





**Table 5.** Parametric table Arnold et al. (2012a)

| Parameter | Full meaning |
|---|---|
| ESCO.hru | Soil evaporation compensation factor |
| SOL_AWC.sol | Available water capacity of the layer of soil |
| CH_N2.rte | Mannings 'n' coefficient |
| SURLAG.bsn | Surface runoff lag coefficient |
| GW_DELAY.gw | Groundwater delay-time |
| GWQMN.gw | Groundwater minimum threshold |
| OV_N.hru | Mannings 'n' value for overland flow |
| GW_REVAP.gw | Groundwater revap coefficient |
| RCHRG_DP.gw | Deep aquifer percolation fraction |
| SOL_BD.sol | Moist bulk density |
| CH_K2.rte | Effective hydraulic conductivity in the main alluvium channel |
| CN2.mgt | Curve number |
| SOL_K.sol | Saturated hydraulic conductivity |
| ALPHA_BF.gw | Baseflow alpha factor |





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
