# Peer review of "Hydro-Climatic Modelling of an Ungauged Basin in Kumasi, Ghana"

_Hydrology and Earth System Sciences, 2017_

## Referee Comment (RC1) · Anonymous Referee #1 · 22 Jan 2018

**OVERVIEW**

The manuscript investigates the impact of climate changing on the water resources management of the Owabi catchment in Ghana. Specifically, the SWAT hydrological model was calibrated with past streamflow observations (from a neighbouring basin) and then used for simulating the different component of the hydrological cycle (precipitation, evapotranspiration, infiltration, . . .) for past and future climate conditions (obtained by GCM for different scenarios).

**GENERAL COMMENTS**

The manuscript investigates the impact of climate changing in an ungauged basin in Ghana. As the study area is in a developing country, with missing data, the obtained results are surely of general interest. However, the paper fails in showing new scientific results and the scientific soundness should be improved.

I listed below several major comments that should be addressed to make the paper results more robust and significant. Additionally, I would suggest converting the paper as Technical Note as it seems to me more appropriate.

1) The paper uses a single GCM. The uncertainty of this choice might be significantly large. What are the results for different GCMs? Do they go in the same direction? This must be tested.

2) The analysis is carried out for a very small basin (70 kmq). The spatial scale of GCM is much larger. It introduces further uncertainties that should be considered and discussed. For instance, the use of Regional Climate Models might be needed. As before, it must be investigated and assessed.

3) Discharge observations from a neighbouring basin are used for SWAT model calibration and validation. However, no details are given on the used dataset, and how it is used as a proxy of discharge observations for the investigated basin. It should be

clarified and clearly specified.

4) Overall, several details are missing in the explanation of the methodology. For instance, two different land use scenarios are considered but no information is given on the criteria used for the selection of such scenarios. Similarly, for the selection of a single GCM. A single hydrological model. A single downscaling approach (not explained properly). All these aspects must be clarified.

5) SWAT model is calibrated with discharge observations only (and not for the same basin). In the paper it is speculated what are the different contribution of precipitation and evapotranspiration on surface and subsurface runoff. By using only discharge data for model calibration, it is not possible to give information on the different components of the hydrological cycle. Different SWAT parameterizations might provide the same performances in terms of discharge simulation while providing very different shares in the hydrological components. The overall discussion should be removed and totally revised.

Several specific comments and corrections should be also addressed. However, I believe the paper in the current form should be significantly modified and, hence, I have not included the specific corrections at this stage.

I understand that the review is not positive. However, I believe there is a strong potential in the performed study as we surely need to investigate what will be the impact of climate changing in developing countries, and this study might provide an important contribution in this respect. Based on the above comments, I believe the paper needs a major revision before to be re-evaluated for its technical content.

---

## Author Comment (AC1) · 29 Jan 2018

The authors would like to thank our anonymous reviewer for his constructive comments.

1) The paper uses a single GCM. The uncertainty of this choice might be significantly large. What are the results for different GCMs? Do they go in the same direction? This must be tested.

RESPONSE: The study used projection data from the Canadian Regional Climate Model of the Fourth Generation (CanRCM4), which is a regional climate model and not a GCM.

2) The analysis is carried out for a very small basin (70 kmq). The spatial scale of GCM is much larger. It introduces further uncertainties that should be considered and

discussed. For instance, the use of Regional Climate Models might be needed. As before, it must be investigated and assessed

RESPONSE: As stated in earlier, the Canadian Regional Climate Model (CanRCM4) was used and this is clearly stated on page 7 line 8, with the data being bias corrected before use. The spatial resolution is quite coarse and hence, the manuscript is being updated with a new projection at 0.22 lat/lon (approximately 25 km) and three different ensembles for the various RCPs to observe the future discharge trends.

3) Discharge observations from a neighbouring basin are used for SWAT model calibration and validation. However, no details are given on the used dataset, and how it is used as a proxy of discharge observations for the investigated basin. It should be clarified and clearly specified.

RESPONSE: The study used monthly streamflow data from the nearest catchment, the Offin Basin and spanned a period of 2001 to 2010. This information is seen in table 1. However, it has again been updated in the text. Please note that, the data was not subjected to any rigorous statistical tests before use, however, we applied the spatial proximity global arithmetic mean method (page 6 line 8-9) as used in Oudin et al. (2008) and the results obtained with the raw data for calibration purposes were good for the Owabi catchment.

4) Overall, several details are missing in the explanation of the methodology. For instance, two different land use scenarios are considered but no information is given on the criteria used for the selection of such scenarios. Similarly, for the selection of a single GCM. A single hydrological model. A single downscaling approach (not explained properly). All these aspects must be clarified.

RESPONSE: The following has been updated in the manuscript accordingly:

(i) Criteria for landuse scenario: Although the catchment is prone to anthropogenic invasion, there have been strict forestry rules that safe-guards the forest from human

occupation. These rules are unlikely to be relaxed for the benefit of the residents since the catchment is prided as an international and only inland ramsar site in Ghana. Forest guards are observed to patrol the forest at least twice a day, with severe penalties awarded on any defaulters. It is therefore likely, that current settlement areas can either remain the same, or grasslands and croplands would be urbanised to support growing population. These assumptions formed the criteria for developing the two landuse scenarios.

(ii) As previously stated, the Canadian Regional Model of the fourth generation (Can-RCM4) was used.

(iii) The SWAT model was chosen for its thorough modelling of hydrology of catchment as observed in most hydrological manuscript. On the other hand, this study was undertaken through the Building Stronger Universities Phase 2 (BSU II) project, where panelists selected the SWAT model to be most convenient for assessing the Owabi catchment hydrology.

(iii) The CmHyd software has about seven bias correction options available for precipitation and temperature. This included; distribution mapping of precipitation and temperature, linear scaling, delta-change correction, precipitation local intensity scaling, power transformation of precipitation. All these options were used to correct for biases in the historic rainfall and temperature datasets which included the observed data and that of the RCM for different RCP scenarios. After correction, the best bias correction option was the distribution mapping and hence was chosen for the hydrological climate change analysis. This option has also been found to be the most reliable per Teutschbein and Seibert (2012).

(5) SWAT model is calibrated with discharge observations only (and not for the same basin). In the paper it is speculated what are the different contribution of precipitation and evapotranspiration on surface and subsurface runoff. By using only discharge data for model calibration, it is not possible to give information on the different components

of the hydrological cycle. Different SWAT parameterizations might provide the same performances in terms of discharge simulation while providing very different shares in the hydrological components. The overall discussion should be removed and totally revised.

RESPONSE: It is certain that the calibrated model was based on discharge only. With the exception of precipitation (which is unlikely to undergo calibration), records for the other components of the water balance are non-existent at the catchment and hence cannot be calibrated. The discharge calibrated model therefore served as a first check to assess the degree to which the water balance reflected that of the catchment area. A second check was done using the SWAT Check tool to analyse any errors with the monthly averaged values of the water balance components. This also proved successful, as no errors were identified, taking into account the soil and landuse categories within the catchment.

References: Oudin, L., Andréassian, V., Perrin, C., Michel, C., and Le Moine, N.: Spatial proximity, physical similarity, regression and ungaged catchments: A comparison of regionalization approaches based on 913 French catchments, Water Resources Research, 44, 2008.

Teutschbein, C. and Seibert, J.: Bias correction of regional climate model simulations for hydrological climate-change impact studies: Review and evaluation of different methods, Journal of Hydrology, 456, 12–29, 2012.

---

## Referee Comment (RC2) · Anonymous Referee #2 · 12 Feb 2018

**Comments on "Hydro-Climatic Modelling of an Ungauged Basin in Kumasi, Ghana"**

**Overview:**

This manuscript presents on an important topic of water balance processes and their changes over time according to climate change in an ungauged basin. This article focused on the Owabi catchment located in the Ghana and which provides about 20% of drinking water demand of the Kumasi metropolis. The aims of this study are: 1) To simulate stream-flow and establish the water balance for the catchment thanks to the SWAT model; for this purpose, a sensitivity, calibration and validation analyzes were performed using SWAT-CUP. 2) To forecast the streamflow for the catchment under three different climatic scenarios and two land-use changes.

**General comments:**

I am glad to see the many improvements on the paper since the last submission. I think this new work has a good publication potential, however, more work needs to be done to improve the process and the conclusions. Based on the comments below, I think this paper needs a major revision before a re-evaluation for publication.

1) The purpose of your manuscript indicates that you will establish the water balance of the watershed. However, only Figure 5 shows a water balance, not complete elsewhere, which is only a monthly average. It would have been interesting to make an assessment, even annual, on the different years of simulation (calibration + validation) to get an idea of the different terms and their evolution over time. In addition, a complete analysis would make it possible to check that the water balance is closed. It would also be interesting to add in the total water balance, the surface, lateral and groundwater contribution part instead of the water yield. The definition of this last term must be modified on page 6 line 14: WYLD = SURQ + LATQ + GWQ -TLOSS - pond abstractions. Using only discharge data to explain the different contribution of precipitation and evapotranspiration on total runoff is not enough.

2) Many details are missing in the methodology section and in particular explanations about the choices that the authors have done. Some examples: Why this partial evolution of vegetation cover? How precisely climate biases have been corrected? How are the comparisons made on the different scenarios? Which criterion is used in this case? Where are located the meteorological stations? (Closest is not sufficient, gives values and indicates on figure), Why did you use the Hargreaves method for evapotranspiration? (just temperature data are available?) This information should also appear in the methodology section and not only in the result section. Check all this part.

3) The observed discharge from the neighboring station, Offin river are used for this study. The regionalization method to determine runoff is a strong assumption in an article on calibration and validation of a model from these data. It seems important to me that the authors develop this part and explain in detail the different steps, datasets used etc. According to Hrachowitz et al. (2013), to regionalize, it is necessary to

admit a homogeneity in terms of land use, topography and land cover between watersheds, is that true here? Expend this part.

4)    The authors list the different uncertainties associated with the observation data but do not show their impact on the results. It would be interesting to add an "uncertainty" part to the input data regardless of the 99PPU SWAT-CUP analysis.

5)    A first calibration step is performed on various sensitive parameters of the model (9): CN2, SURLAG, ESCO, SOL_BD, SOL_AWC, CH_N2, ALPHA_BF, RCHRG_DP, and GW_REVAP. The calibrated model is then used as to generate runoff with the different landscape scenarios.  However, calibration of parameters such as CN2 or SOL_AWC is not possible if thereafter, the surface condition changes. The authors must think carefully about the parameters that can be used in the calibration without biasing the scenario part. The authors should be careful with the calibration of some parameters taking into account the possible bias on the scenarios results.

6)    Some parts of the article should be changed or even deleted to focus on a more innovative and interesting part to clearly see the originality of the paper (I suggest you read Srinivasan et al., 2010 or Sisay et al., 2017 for example).

**Technical and specific comments:**

I listed few important technical comments here but I have not developed this part at this stage.

Figure 1: Keep the same term: "Catchment" or "Watershed"

Figure 2: Add the hydro-Meteorological stations on this figure

Figure 3: Add "(LU1)" in the legend as in the Figure 4

Figure 9 and 10: Not readable, zoom over hydrological year

Table 1: Add "Temporal resolution" or "Acquisition dates" for all data (DEM etc.)

Page 2, Line 6: Management

Page 3, Line 54: Keep the same precision of the surface area (69.72 here and 69 in the abstract)

**References:**

Sisay, E., Halefom, A., Khare, D., Singh, L. and Worku, T.: Hydrological modelling of ungauged urban watershed using SWAT model, Model. Earth Syst. Environ., 3(2), 693, doi:10.1007/s40808-017-0328-6, 2017.

Srinivasan, R., Zhang, X. and Arnold, J.: SWAT ungauged: Hydrological budget and crop yield predictions in the upper mississippi river basin, Am. Soc. Agric. Biol. Eng., 53(5), 1533–1546, 2010.

---

## Author Comment (AC2) · 23 Feb 2018

The authors would like to express gratitude to our anonymous reviewer for taking the time to review our manuscript again. We appreciate his constructive comments very much.
Response to comments are shown in 'bold'.

1) The purpose of your manuscript indicates that you will establish the water balance of the watershed. However, only Figure 5 shows a water balance, not complete elsewhere, which is only a monthly average. It would have been interesting to make an assessment, even annual, on the different years of simulation (calibration + validation) to get an idea of the different terms and their evolution over time. In addition, a complete analysis would make it possible to check that the water balance is closed. It would also be interesting to add in the total water balance, the surface, lateral and groundwater contribution part instead of the water yield. The definition of this last term must be modified on page 6 line 14: WYLD = SURQ + LATQ + GWQ -TLOSS - pond abstractions. Using only discharge data to explain the different contribution of precipitation and evapotranspiration on total runoff is not enough.

**Two new figures have been plotted and updated for annual evolution of the water balance from 1986 to 2015 in the manuscript. The first new figure shows the annual water balance comprising of rainfall, surface runoff, soil moisture, actual evapotranspiration and potential evapotranspiration. Groundwater, lateral and percolated water flows had low values and hence were plotted on a different figure to show their trends. The water yield equation has been updated according as well as appropriate discussions in the water balance section of the paper.**

2) Many details are missing in the methodology section and in particular explanations about the choices that the authors have done. Some examples: Why this partial evolution of vegetation cover? How precisely climate biases have been corrected? How are the comparisons made on the different scenarios? Which criterion is used in this case? Where are located the meteorological stations? (Closest is not sufficient, gives values and indicates on figure), Why did you use the Hargreaves method for evapotranspiration? (just temperature data are available?) This information should also appear in the methodology section and not only in the result section. Check all this part.

**Choice/criterion for the evolution of vegetation cover: Although the catchment is prone to anthropogenic invasion, there have been strict forestry rules that safe-guards the forest from human occupation. These rules are unlikely to be relaxed for the benefit of the residents since the catchment is prided as an international and only inland ramsar site in Ghana. Forest guards are observed to patrol the forest atleast twice a day, with severe penalties awarded on any defaulters. It is therefore likely, that current settlement areas can either remain the same, or grasslands and croplands would be urbanised to support growing population. These assumptions formed the criteria for developing the two landuse scenarios or the evolution of the vegetation cover. Comparison between the scenarios was done based on streamflow to ascertain the amounts that would be available in future for water processing, demand and supply within the Kumasi metropolis. In the study, only one climate ensemble was used for the projections, however we have now used three different climate ensembles under each of the different RCPs for the future projections at a higher resolution of 0.22 lon/lat, as against the 0.44 lon/lat used currently in the study. This will give a much clearer view on the future dynamics of streamflow under various climatic conditions.**

**The CmHyd software has about seven bias correction options available for precipitation and temperature. This included; distribution mapping of precipitation and temperature, linear scaling, delta-change correction, precipitation local intensity scaling, power transformation of precipitation. All these options were used to correct for biases in the projection rainfall and temperature datasets with reference to the observed rainfall and temperature data which also served as an input. After correction, the best bias correction option was the distribution mapping and hence was chosen for the hydrological climate change analysis. This option has also been found to be the most reliable per Teutschbein and Seibert, 2012.**

**The Offin, Barekese and Kumasi airport meteorological stations used for data in-filling for Owabi are not located within the catchment area. Hence to provide clarity of the image of the study area (Fig1), it is impossible to show these stations on the map. However, the study area map has been updated to show the Owabi meteorological station, whiles the coordinates of the other stations have been stated in the 'hydrometeorological data' section to give clearer understanding of their distance from the Owabi catchment.**

**The Hargreaves method was used for the calculation of evapotranspiration because only rainfall and temperature datasets were available at the study area. This information has been inserted in the methodology.**

3) The observed discharge from the neighboring station, Offin river are used for this study. The regionalization method to determine runoff is a strong assumption in an article on calibration and validation of a model from these data. It seems important to me that the authors develop this part and explain in detail the different steps, datasets used etc. According to Hrachowitz et al. (2013), to regionalize, it is necessary to admit a homogeneity in terms of land use, topography and land cover between watersheds, is that true here? Expend this part.

**The study used monthly streamflow data from the nearest catchment, the Offin Basin and spanned a period of 2001 to 2010. This information is seen in table 1. However, it has again been updated in the text. Please note that, the data was not subjected to any rigorous statistical tests before use, however, we applied the spatial proximity global arithmetic mean method (page 6 line 8-9) as used in Oudin et al. (2008) and the results obtained with the raw data for calibration purposes were good for the Owabi catchment. Again, the spatial proximity does not rely heavily on physical characteristics of the  watersheds but instead on a dense gauge network, but it is worth knowing that both catchments (Owabi and Offin) are dominated by the same soil characteristics, but the landuse dynamics and topography are slightly different.**

4) The authors list the different uncertainties associated with the observation data but do not show their impact on the results. It would be interesting to add an "uncertainty" part to the input data regardless of the 99PPU SWAT-CUP analysis.

**The uncertainty analysis has been rediscussed as much as possible to reflect their impact on the results obtained during calibration and validation. These would be found in the revised manuscript.**

5) A first calibration step is performed on various sensitive parameters of the model (9): CN2, SURLAG, ESCO, SOL_BD, SOL_AWC, CH_N2, ALPHA_BF, RCHRG_DP, and GW_REVAP. The calibrated model is then used as to generate runoff with the different landscape scenarios. However, calibration of parameters such as CN2 or SOL_AWC is not possible if thereafter, the surface condition changes. The authors must think carefully about the parameters that can be used in the calibration without biasing the scenario part. The authors should be careful with the calibration of some parameters taking into account the possible bias on the scenarios results.

**Upon a second review, it was noticed that the model was highly overparameterised and ill-conditioned. For instance, the parameter value of CN2 was out of range after calibration. The model calibration has been rerun and new sensitive values include CN2, ALPHA_BF and SURLAG, with the other parameters being highly insensitive. As regarding the hydrological soil group "D" at the catchment, it is not surprising that most groundwater parameters were not sensitive such as the delay time, recharge depth among others. The D soil group is characterised by low infiltration and high surface runoff capacity. Therefore, we assume that these new sensitive parameters illustrate the hydrology of the catchment and result in the reduction of biases which might be introduced into future projections.**

6) Some parts of the article should be changed or even deleted to focus on a more innovative and interesting part to clearly see the originality of the paper (I suggest you read Srinivasan et al., 2010 or Sisay et al., 2017 for example).

**The entire manuscript would be reviewed by authors and appropriate redundant sections removed. The suggested manuscripts (Srinivasan et al., 2010 and Sisay et al., 2017) would also be reviewed to boost our paper.**

Technical and specific comments:
I listed few important technical comments here but I have not developed this part at this stage.

Figure 1: Keep the same term: "Catchment" or "Watershed"

**The word "Catchment' would be used throughout the manuscript.**

Figure 2: Add the hydro-Meteorological stations on this figure

**For clarity of the study map, only the Owabi meteorological station have been added to the map but the geographical coordinates of the other stations have been stated in the 'hydrometeorological data' section.**

Figure 3: Add "(LU1)" in the legend as in the Figure 4

**LU1 has been updated in figure 3.**

Figure 9 and 10: Not readable, zoom over hydrological year

**There will be new figures to replace figures 9 and 10 for the streamflow projections and the images made clearer in the manuscript update.**

Table 1: Add "Temporal resolution" or "Acquisition dates" for all data (DEM etc.)

**Temporal resolutions have been added to the DEM (2000-02-01:2000-02-29), Soil map (2007-02-28) in table 1.**

Page 2, Line 6: Management

**Management has been updated.**

Page 3, Line 54: Keep the same precision of the surface area (69.72 here and 69 in the abstract)

**The surface area of "69 km2", will be used throughout the manuscript.**

**REFERENCES**
**Oudin, L., Andréassian, V., Perrin, C., Michel, C., and Le Moine, N.: Spatial proximity, physical similarity, regression and ungaged catchments: A comparison of regionalization approaches based on 913 French catchments, Water Resources Research, 44, 2008.**

**Teutschbein, C. and Seibert, J.: Bias correction of regional climate model simulations for hydrological climate-change impact studies: Review and evaluation of different methods, Journal of Hydrology, 456, 12–29, 2012.**

---

## Referee Comment (RC3) · Anonymous Referee #3 · 17 Mar 2018

**Hydro-Climatic Modelling of an Engaged Basin in Kumasi, Ghana.**

This manuscript presents an application of the SWAT hydrological model on the Owabi basin in Ghana. Once calibrated, the model was fed with climate change scenarios to assess the future availability of water on the basin. In addition, two hypothetic future land-use maps where used in combination with climate scenarios, to also assess the impact of a change in land-use on water resources. The first land use map corresponds to the current observed land-use map in the basin, thus assuming no evolution of the land use over time. The second land use map assumes extensive urbanisation of the basin in the future. The authors explain that as such, the Owabi basin is ungauged, the closest streamflow gauge being located 11 km away.

In my opinion, this manuscript is unfortunately not appropriate for publication in HESS and should be rejected. There are several reasons for that, the most important being that I find the level of originality and innovation of the work to be quite low. I would encourage the authors to submit a revised version of their manuscript to a smaller journal specialized in applications and/or case studies.

*Major comments:*

1.  The originality of the work is low

In my opinion, this manuscript presents the application of a well known model to a new catchment. As such, there is nothing new with that. Commonly available data with very little pre-processing, as well as a standard calibration process were used. Readily available weather data for climate change scenarios were used, again with no pre-processing. There are already several papers in the literature focused on using hydrological models with climate scenarios to project future water availability.

I encourage the authors to follow the scientific method: what do you want to verify? State your hypothesis! What/where is your contribution? I advise mentioning explicitly in the introduction what your contribution to the advancement of science is.

2.  Some methodological/conceptual elements need clarification

•   The Owabi basin is ungauged. To me, this is a central issue for this work, but only very little information is provided in the manuscript about that. For one thing, although it is mentioned that the nearest gauging station is located 11 km away from the Owabi basin, on another basin called  Offin. a map showing its location is not provided. Such a map would be very useful. In addition, information about the similarities and differences between those two basins (area, slopes, vegetation, soil, etc) should be provided. Finally, although you are using an existing method for transferring streamflow from Offin to Owabi, I believe you should nonetheless describe the method succinctly. Much more emphasis should be given throughout the manuscript to the fact that the basin is ungauged and that the calibration of SWAT was thus performed using streamflow (and corresponding meteorological observations, I would assume) from another basin.

•   To me, although you mention bias correction for the climatic projections on page 7 line 10, it is not clear how the quality of those simulations was assessed. I would have liked to see much more details regarding those biases (with graphs and numbers) and also more details about the efficiency of the bias removal process. In addition, although you mention on page 7 that « These were projected under three Representative Concentration Pathways (…) »

you do not provide any detail about what those projected scenarios represent (for instance, which one is worst than the other in terms of greenhouse gaz emissions). I would have appreciated more details.

- You also mention on line 12, page 7, that distribution mapping assumes a gaussian distribution of the dataset. Have you verified that the datasets indeed follow gaussian distributions? Again, more details regarding that, alone with test results, should be provided. Why using two bias correction methods instead of one? Can you provide comparative graphs of the outcomes of those methods? How do they compare?

- I do not find Appendix A to be useful, as it contains basic equations that are very well-known in hydrology. I would advise removing it.

- The SWAT model operates on a daily time scale but you analyse data at a monthly time scale. I also understand that you might have access only to monthly streamflow observations for the Offin basin. I would have appreciated much more detail about the choice of the monthly timescale and how it impacts model calibration and data preparation, as well as post-processing of the outputs (simulated streamflows).
- The authors mention an uncertainty enveloppe on simulated streamflow. This envelope is not displayed on figures, and the methodology to compute it is also not explained. Again, more explanations are needed. The reader can only guess that it was obtained through the use of SWAT-CUP (page 8).

*Specific comments*

1. Typos/spelling mistakes: I would advise a very thorough linguistic revision. These are only a few examples:
   - The word « streamflow » is sometimes spelled « stream-flow ». Please check the entire manuscript and ensure consistency.
   - Page 1, line 15, page 8 line 15, page 10 line 3 and several other instances: please replace « at the catchment » by « at the catchment scale » or « on the catchment », depending on what you mean.
   - Page 2 line 28: on the Bani… (not at)
   - Page 2 line 30: « but sometimes highly overestimated… »
   - Page 3 line 2: « conclusion IS given in section 4. »
   - Page 3 line 5: Correct « … in Ghana. redIt comprises of the forest »
   - Page 8 line 15: remove comma after « that »
   - Page 8 line 16: correct « … soil group of type D… »
   - Page 9 line 15: correct « …the root nature of the forested trees …». Perhaps by « the forested area » or « trees in the forest ».
   - Page 10 line 21: « The catchment topography ranged from … » should be corrected to « The catchment topography ranges from … »
   - Page 14 line 3: Replace « welcoming » by « welcome »
   - Page 15 line 13: Replace « projection » by « projected »
   - Page 16 line 2-3: The sentence « Specifically… for the catchment » is confusing and needs rephrasing. As written now, I understand that the model can « simulate its own calibration ».

2. Page 1 line 2: « Major stakeholders… » please add a reference.
3. Page 3 line 8: There is a problem with one reference « Commission » is not a valid author name.

4. Page 4 after equation (1): « t » should be in italics. Also, there is a sequence in which you name 5 symbols but only mention 4 definitions (« Rday, Qsurf… »)
5. Page 6 line 6: « … it has been revealed that regionalization and other genetic networks… » I don't understand what you mean by genetic network. Please clarify.
6. Page 8 line 1: A reference is needed for the SUFI-2 algorithm.
7. Page 8 line 12: I suggest nuancing the sentence « Unlike rainfall that is easily measurable… ». There are lots of issues regarding accurate rainfall measurement (for instance under-captation and lack of spatial coverage of ground stations).
8. Page 10 line 15: You say that 14 parameters have been selected from literature. First, you need to cite some supporting references. In addition, again, this is much too vague. Why are those parameters so important compared to others?
9. Page 10 line 21: Given its importance, the long name of parameter CN2 should be mentioned  (even if it is in the Appendix). In fact, this table should be in the text and not in an appendix. Incidentally the number of the Appendix as referred to in the text is not good (it is table 5, not Appendix 5).

---

## Author Comment (AC3) · 22 Mar 2018

The authors would like to thank our anonymous reviewer for his immense review and criticisms. We here provide responses to his comments. Responses are written in **bold italics** and comments in plain text**.**

1. The originality of the work is low. In my opinion, this manuscript presents the application of a well known model to a new catchment. As such, there is nothing new with that. Commonly available data with very little pre-processing, as well as a standard calibration process were used. Readily available weather data for climate change scenarios were used, again with no pre-processing. There are already several papers in the literature focused on using hydrological models with climate scenarios to project future water availability. I encourage the authors to follow the scientific method: what do you want to verify? State your hypothesis! What/where is your contribution? I advise mentioning explicitly in the introduction what your contribution to the advancement of science is.

***The objective of the paper would be made explicitly clearer in the introduction of the updated manuscript. A major component which was not adequately highlighted was the fact that the Owabi catchment is ungauged and how the chosen regionalization method for the transfer of stream-flow data from the Offin basin to the Owabi catchment impacts the calibration results. The spatial proximity method which was chosen as the data transfer scheme relies on a dense gauge network in the vicinity of the proposed catchment. Most studies have also applied this scheme for transferring data onto larger catchment area. It was therefore hypothesized how the spatial proximity scheme can be used on a smaller catchment using only one proxy station (Offin basin) as donor. These information would be updated accordingly.***

2. The Owabi basin is ungauged. To me, this is a central issue for this work, but only very little information is provided in the manuscript about that. For one thing, although it is mentioned that the nearest gauging station is located 11 km away from the Owabi basin, on another basin called Offin. a map showing its location is not provided. Such a map would be very useful. In addition, information about the similarities and differences between those two basins (area, slopes, vegetation, soil, etc) should be provided. Finally, although you are using an existing method for transferring streamflow from Offin to Owabi, I believe you should nonetheless describe the method succinctly. Much more emphasis should be given throughout the manuscript to the fact that the basin is ungauged and that the calibration of SWAT was thus performed using streamflow (and corresponding meteorological observations, I would assume) from another basin.

***The gauging station at Offin is located 11 km from the Barekese dam. This dam is also found 19 km away from the Owabi catchment. This makes a total distance of about 30 km between the Owabi catchment and the Offin basin gauging station. This has been updated in the manuscript. For clarity of the map and owing to the distance apart, only the Owabi catchment alongside its neighbouring towns can be projected on the study map. Mean elevation at Offin is about 277m as opposed to 265 m at the Owabi catchment. They both have the same soils (orthic acrisols), semi-decidous forest types, rapid urbanisation within the last decade as well as influenced by bi-modal rainfall regimes. Geologically, the Owabi catchment falls within the Birimian meta-sediment of the Kumasi Basin which consists of phyllites, granodiorites, schists, greywackes, tuffs with its associated granitoid, whiles the Offin Basin is underlain by the Voltain, Birimian and Granite rock types. Although the hydrometeorological stations cannot be available on the map, their geographical coordinates have been stated in the 'hydrometeorological data' section.***

***We applied the spatial proximity global arithmetic mean method (page 6 line 8-9) as used in Oudin et al. (2008), which relies mainly on a dense gauge network. However, the results obtained with only***

*one hydrological station data yielded acceptable results during calibration for the Owabi catchment. The Owabi catchment has a meteorological station situated on its premises, the other meteorological station data used were only to fill in gaps which existed in some years in the Owabi data. We believe that emphasis was placed on the use of the Offin data as the source of stream-flow data for the Owabi catchment in table 1. However, this has been emphasized in appropriate areas of the text.*

To me, although you mention bias correction for the climatic projections on page 7 line 10, it is not clear how the quality of those simulations was assessed. I would have liked to see much more details regarding those biases (with graphs and numbers) and also more details about the efficiency of the bias removal process. In addition, although you mention on page 7 that « These were projected under three Representative Concentration Pathways (...) » you do not provide any detail about what those projected scenarios represent (for instance, which one is worst than the other in terms of greenhouse gaz emissions). I would have appreciated more details.

*The CmHyd software has about seven bias correction options available for precipitation and temperature. This included; distribution mapping of precipitation and temperature, linear scaling, delta-change correction, precipitation local intensity scaling, power transformation of precipitation. All these options were used to correct for biases in the historic rainfall and temperature datasets which included the observed data and that of the RCM for different RCP scenarios. After correction, the best bias correction option was the distribution mapping which simulated historic rainfall and temperature patterns as close as possible to the observed historic trends. This option was therefore chosen for the climate change analysis. Figures such as that found below, showing trends in the mean monthly rainfall/temperature as well as the performance of the method would be included in the revised manuscript. To assess the quality of simulations, the figures showing the coefficient of variation and standard deviations would be provided. Further details would also be given on the Representative Concentration Pathways in the manuscript.*

**Figure A**

[Figure]

[Figure]

*Distribution mapping correction method of rainfall (RCP2.6): Comparison of observed (p31022_ovl) model historic without correction (PCP0001_ovl), and model historic corrected (PCP0001_ovl_dm_hist).*

**Figure B**

[Figure]

*Distribution mapping correction method of temperature [Tmax and Tmin] (RCP2.6): Comparison of observed (t31022_ovl_max/min) model historic without correction (TMP_0001_ovl_max/min), and model historic corrected (TMP_0001_ovl_dm_hist_max/min).*

You also mention on line 12, page 7, that distribution mapping assumes a gaussian distribution of the dataset. Have you verified that the datasets indeed follow gaussian distributions? Again, more details regarding that, alone with test results, should be provided. Why using two bias correction methods instead of one? Can you provide comparative graphs of the outcomes of those methods? How do they compare?

*The datasets were not verified to follow the gaussian distribution, but contrary to temperature, the high variability of tropical rainfall does not follow the gaussian distribution. Therefore, within the CmHyd software, the gaussian and gamma distributions embedded in the distribution mapping of precipitation and temperature modules are used to correct the temperature and rainfall datasets respectively. The module creates a transfer function which corrects the distribution of historical rainfall and temperature to the observed measurements. The working principle of the module is such that, with rainfall, a gamma distribution is assumed with a shape parameter ($\alpha$) and a scale parameter ($\beta$). Alpha controls the distribution profile, such that, when $\alpha < 1$, the Gamma distribution is exponentially shaped, $\alpha = 1$ is a special case and characterizes an exponential distribution whiles $\alpha > 1$ shapes a skewed uni-modal distribution curve. The scale parameter $\beta$ on the other hand, determines the dispersion of the Gamma distribution with a smaller $\beta$ leading to a more compressed distribution and a lower probability of extreme events and a larger $\beta$, causing a stretched distribution, with higher probabilities of extreme events.*

*For temperature time series, the Gaussian distribution with location parameter ($\mu$) and scale parameter $\sigma$ is assumed to give the best fit. The scale parameter $\sigma$ determines the standard deviation of the Gaussian distribution and smaller values imply a more compressed distribution with lower probabilities of extreme values. Contrary, a larger value for $\sigma$ indicates a stretched shape with higher probabilities of extreme values. The location parameter $\mu$ directly controls the mean and therefore, the location of the distribution. Further details can be found in the Teutschbein and Seibert (2012). These details has been be added to the manuscript along with figures and numbers (where necessary) to show the performance of the bias correcting method.*

I do not find Appendix A to be useful, as it contains basic equations that are very well-known in hydrology. I would advise removing it.

*These equations are indeed well-known but nonetheless very important in stating them for clarity and for useful resource. For this reason, they were not put in the main text but the appendix. The authors therefore think it should be retained.*

The SWAT model operates on a daily time scale but you analyse data at a monthly time scale. I also understand that you might have access only to monthly streamflow observations for the Offin basin. I would have appreciated much more detail about the choice of the monthly timescale and how it impacts model calibration and data preparation, as well as post-processing of the outputs (simulated stream-flows).

*The SWAT model operating on a daily timescale implies that the climatic input variables (rainfall and temperature) are formatted into daily or sub-daily values before being used as input for the model. The model can therefore run on the daily climate variables and give output simulations in either daily totals, monthly or annual means as preferred by the modeller. Due to the availability of monthly observed stream-flow, the monthly simulated streamflow was the best choice. Again, after calibration, the SWAT is known to perform better in simulating monthly than daily stream-flows (Uzeika, 2011), since most daily errors are compensated for on the monthly timescale.*

The authors mention an uncertainty enveloppe on simulated streamflow. This envelope is not displayed on figures, and the methodology to compute it is also not explained. Again, more explanations are needed. The reader can only guess that it was obtained through the use of SWAT-CUP (page 8).

*The following has been updated in the manuscript: The p and r-factors at a 95% prediction uncertainty determine the degree of uncertainty associated with the model. A p-factor of 1 and an r-factor which 0 is usually considered as a perfect model fit with observed with no uncertainty. However, for streamflow, a p-factor >=0.70 is the better choice of model fit, although it is also acceptable for the model to cover more than half of the observed data. These factors are calculated during model calibration and validation within the SWAT-CUP tool.*

*The envelope is not displayed on figures, but the values corresponding to the p and r factors have been shown in Table 4. The gnuplot software was used which is unable to capture the uncertainty envelope, unlike the default MS-Excel-like plot tool which is embedded in SWAT-CUP, however this is being considered in the revised version to capture the envelope. The uncertainty values are also shown in Table 4 to give a fair idea.*

*For your perusal, we have inserted the default plots from SWAT-CUP to show the envelope (in green color) below.*

*(a) Calibration plot*

[Figure]

*(b) Validation plot*

[Figure]

**Specific comments**

1. Typos/spelling mistakes: I would advise a very thorough linguistic revision. These are only a few examples:

• The word « streamflow » is sometimes spelled « stream-flow ». Please check the entire manuscript and ensure consistency.

*For consistency, all spellings within the manuscript has been changed to stream-flow.*

• Page 1, line 15, page 8 line 15, page 10 line 3 and several other instances: please replace « at the catchment » by « at the catchment scale » or « on the catchment », depending on what you mean.

*Page 1, line 15: Catchment replaced with 'at the catchment scale'.*
*Page 8 line 15 and page10 line 3: At the catchment replaced with 'on the catchment'.*

• Page 2 line 28: on the Bani... (not at)

*Word replaced with 'at the Bani'.*

• Page 2 line 30: « but sometimes highly overestimated... »

*Statement modified to, ' but sometimes overestimated ...'*

• Page 3 line 2: « conclusion IS given in section 4. »

*The word 'is' has been inserted.*

• Page 3 line 5: Correct « ... in Ghana. redIt comprises of the forest »

*The word 'redIt' has been modified to 'it'.*

• Page 8 line 15: remove comma after « that »

*The comma has been removed.*

• Page 8 line 16: correct « ... soil group of type D... »

*The statement has been modified.*

• Page 9 line 15: correct « ...the root nature of the forested trees ...». Perhaps by « the forested area » or « trees in the forest ».

*Statement replaced with, 'the forested trees'.*

• Page 10 line 21: « The catchment topography ranged from ... » should be corrected to « The catchment topography ranges from ... »

*The statement has been modified to 'ranges from'.*

• Page 14 line 3: Replace « welcoming » by « welcome »

*'Welcoming' has been replaced by 'welcome'.*

• Page 15 line 13: Replace « projection » by « projected »

*Projection has been replaced by 'projected'*

• Page 16 line 2-3: The sentence «  Specifically... for the catchment  » is confusing and needs rephrasing. As written now, I understand that the model can « simulate its own calibration ».

*The statement has been rephrased as follows, 'Specifically, the model simulated both  historic and projected stream-flow as well as water balance whiles the SUFI-2 algorithm embedded in the SWAT-CUP software was used for model calibration and validation for the catchment.'*

2. Page 1 line 2: « Major stakeholders... » please add a reference.

*Appropriate references have been added.*

3. Page 3 line 8: There is a problem with one reference « Commission » is not a valid author name.

*The citation has been updated to 'Forestry Commission of Ghana'.*

4. Page 4 after equation (1): « t » should be in italics. Also, there is a sequence in which you name 5 symbols but only mention 4 definitions (« Rday, Qsurf... »)

*The 't' has been italized and the definition of w_seep has been updated in the text as the amount of water entering the vadose zone from the soil profile.*

5. Page 6 line 6: « ... it has been revealed that regionalization and other genetic networks... » I don't understand what you mean by genetic network. Please clarify.

*'Genetic network', is analogous to 'artificial neural networks'. This has been updated in the text.*

6. Page 8 line 1: A reference is needed for the SUFI-2 algorithm.

*A reference has been given as Abbaspour (2015).*

7. Page 8 line 12: I suggest nuancing the sentence «  Unlike rainfall that is easily measurable...  ». There are lots of issues regarding accurate rainfall measurement (for instance under-captation and lack of spatial coverage of ground stations).

*There are undeniable issues regarding the accuracy of rainfall, but the statement used in the text was to imply that, rainfall is the most measurable quantity of the water balance globally and hence can easily to be used as input in hydrological models to simulate the other components of the water balance.*

8. Page 10 line 15: You say that 14 parameters have been selected from literature. First, you need to cite some supporting references. In addition, again, this is much too vague. Why are those parameters so important compared to others?

*A citation has been added, Arnold (2012), which highlights the rate for which these parameters have been used for calibration. The criteria for selection was to ensure that all hydrological processes occurring within the Owabi catchment would be incorporated as much as possible to give the best calibrated result.*

9. Page 10 line 21: Given its importance, the long name of parameter CN2 should be mentioned (even if it is in the Appendix). In fact, this table should be in the text and not in an appendix. Incidentally the number of the Appendix as referred to in the text is not good (it is table 5, not Appendix 5).

*The abbreviation has been written in full on page 10 line 21. The table has also been been inserted in the text and properly referenced.*

**REFERENCES**

Abbaspour, K.: SWAT-CUP, eawag, 2015.

Arnold, J., Moriasi, D., Gassman, P., Abbaspour, K., White, M., Srinivasan, R., Santhi, C., Harmel, R., van Griensven, A., van Liew, M., Kannan, N., and Jha, M.: SWAT:Model use, calibration and validation, Transactions of ASABE, 55, 1491–1508, 2012.

Teutschbein, C. and Seibert, J.: Bias correction of regional climate model simulations for hydrological climate-change impact studies: Review and evaluation of different methods, Journal of Hydrology, 456, 12–29, 2012.

Uzeika, T., Merten, G., Minella, J., and Moro, M.: Use of the SWAT model for hydro-sedimentologic simulation in a small rural watershed, Revista Brasileira de Ciência do Solo, 36, 557–565, 2011.